# Design and Validation of a Multimodal Wearable Device for Simultaneous Collection of Electrocardiogram, Electromyogram, and Electrodermal Activity

**DOI:** 10.3390/s22228851

**Published:** 2022-11-16

**Authors:** Riley McNaboe, Luke Beardslee, Youngsun Kong, Brittany N. Smith, I-Ping Chen, Hugo F. Posada-Quintero, Ki H. Chon

**Affiliations:** 1Biomedical Engineering Department, University of Connecticut, Storrs, CT 06269, USA; 2Institute for Electronics and Nanotechnology, Georgia Institute of Technology, Atlanta, GA 30332, USA; 3Department of Oral Health and Diagnostic Sciences, School of Dental Medicine, University of Connecticut Health, Farmington, CT 06030, USA

**Keywords:** electrodermal activity, electrocardiogram, electromyogram, time domain, frequency domain, wearable devices

## Abstract

Bio-signals are being increasingly used for the assessment of pathophysiological conditions including pain, stress, fatigue, and anxiety. For some approaches, a single signal is not sufficient to provide a comprehensive diagnosis; however, there is a growing consensus that multimodal approaches allow higher sensitivity and specificity. For instance, in visceral pain subjects, the autonomic activation can be inferred using electrodermal activity (EDA) and heart rate variability derived from the electrocardiogram (ECG), but including the muscle activation detected from the surface electromyogram (sEMG) can better differentiate the disease that causes the pain. There is no wearable device commercially capable of collecting these three signals simultaneously. This paper presents the validation of a novel multimodal low profile wearable data acquisition device for the simultaneous collection of EDA, ECG, and sEMG signals. The device was validated by comparing its performance to laboratory-scale reference devices. N = 20 healthy subjects were recruited to participate in a four-stage study that exposed them to an array of cognitive, orthostatic, and muscular stimuli, ensuring the device is sensitive to a range of stressors. Time and frequency domain analyses for all three signals showed significant similarities between our device and the reference devices. Correlation of sEMG metrics ranged from 0.81 to 0.95 and EDA/ECG metrics showed few instances of significant difference in trends between our device and the references. With only minor observed differences, we demonstrated the ability of our device to collect EDA, sEMG, and ECG signals. This device will enable future practical and impactful advances in the field of chronic pain and stress measurement and can confidently be implemented in related studies.

## 1. Introduction

Bioelectrical signals like electrodermal activity (EDA), electrocardiogram (ECG), and surface electromyography (sEMG) can be easily collected noninvasively to assess a range of biophysical conditions including fatigue [1], stress [2], heart diseases [3], and pain [4]. Through novel techniques in signal processing, many signals by themselves have demonstrated the ability to widen such applications. For example, previous studies indicate that the sEMG of various muscles can be a metric of visceral pain which in turn has the ability to differentiate between types of pain such as those with or without temporomandibular disorders [5]. Although monomodal are well suited for some applications, multimodal approaches provide better diagnostic capabilities as physiological systems are not typically affected by one particular signal. In recent years, it has been seen that studies utilizing multiple signals simultaneously can potentially improve the performance of the assessment models by increasing their accuracy and specificity [6]. For example, the impact of the integration of signals has been observed in a study that concluded autonomic dysfunction in visceral pain subjects can be assessed by simultaneously evaluating the sympathetic and parasympathetic functions using EDA and heart rate variability derived from ECG [7]. Others have used multimodal approaches encompassing electroencephalograms (EEG), ECG, EMG, and EDA to analyze cognitive and physical fatigue simultaneously.

Such techniques that utilize simultaneous collection of multiple signals provides greater specificity with respect to isolating the true cause of stress/pain. While one signal may be triggered for a range of stimuli, when superimposed with the unique responses of other signals, false positives can be better eliminated through advanced signal processing and analysis. Most multimodal studies either use separate devices for each signal, many of which are laboratory, large scale devices, or use wearable devices that are only capable of simultaneously collecting one or two of the aforementioned signals [8,9]. This type of approach may hinder the ability to gather required multimodal data due to practical limitations and ultimately cannot be efficiently implemented in a long-term diagnostic setting [10]. To the best of our knowledge, there are no existing multimodal devices available to measure EDA, ECG, and sEMG signals simultaneously. A noninvasive, integrated multi-modal device that allows for simultaneous measurement and analysis of these signals without imposing limitations on a subject could allow us to develop a better understanding of a patient’s pain levels, especially over a long period of time.

EDA is a measurement of the conductance across skin that varies from changes in sweat gland activity. Exosomatic EDA approaches use an external current or voltage to observe the modulated counterpart and produce signals residing in the range of 0.045–0.3 Hz. Such results are typically used to obtain a quantitative measure of sudomotor activity and in turn cognitive arousal/stimulation. ECG captures changes in voltage potential on the skin related to the heart in the form of a distinct, periodic waveform. Depending on the number of electrodes utilized to acquire the signal, ECG can monitor and diagnose various heart conditions. sEMG captures changes in voltage potential either from the skin or subcutaneously that results from the nervous stimulation of muscle. Found within 25–150 Hz, the signal can be analyzed to monitor muscular activity, disease, and regeneration.

In this study, we present the development and validation of a wearable trimodal device that consolidates the hardware required to simultaneously collect EDA, ECG, and sEMG signals using one small chip. While a large portion of the circuitry is based on readily available designs, the integration of the three modules on a single chip maximizes the synchronization of the signal acquisition and the opportunities for practical implementation. The device has the potential to eliminate traditional barriers hindering long term chronic pain signal measurement (fixed devices, superfluous leads, etc.) and dramatically modifies the physical impact such a device would normally have on a subject (greater freedom of movement, unassuming presence, etc.). This wearable can rapidly accelerate research related to the biosignals in question as well as relevant pain and stress studies.

To validate the performance of the integrated trimodal device, we developed a comparison study to observe its performance versus that of commonly utilized industry-standard data acquisition systems. Below, the concept behind our device design as well as its layout and the construction of its circuitry is provided. A description of the comparison study then follows. Each signal module on the device chip correlating to one of the three signals collected had a respective reference device that it was compared to. We obtained various indices from the data collection that allowed for the evaluation of signal quality by analyzing specific characteristics present in the waveforms. These indices computed for both devices’ outputs were compared to ensure similarities between our device and the respective reference. Finally, we performed statistical analysis on the two device data sets when appropriate to provide further insight.

## 2. Materials and Methods

### 2.1. Design and Implementation

#### 2.1.1. Concept Design

The circuit to obtain the trimodal signals (i.e., EDA, sEMG, and ECG) was designed as three separate circuits. The signal processed by each circuit is collected from the subject using dry or gel electrodes. The components selected for the design were chosen because of their performance and the availability of small surface mount packages that make a compact circuit. A simplified overview can be seen in Figure 1a.

#### 2.1.2. Circuitry

The EDA circuit is based on [11]. Briefly, the EDA contains an excitation and detection circuit; the excitation circuit injects current into one electrode and then the resulting signal is detected by the second electrode and circuit. The excitation circuit uses an oscillator circuit (LTC6992, Analog Devices, Wilmington, MA, USA) to generate a 1.5 V peak-to-peak square wave, at a frequency of 100 Hz. A level shifter using a single field effect transistor (FET) is implemented and then the signal is low pass filtered using a Sallen-Key filter with a 3 dB set to 110 Hz [12]. This gives a sine wave output from the excitation electrode. There is a decoupling capacitor in between the circuit output and the excitation electrode. The detection circuit consists of a transimpedance amplifier utilizing an Analog Devices (Wilmington, MA, USA) AD4505 op-amp and an envelope detector to remove additional unwanted signal components.

The ECG circuit is entirely based on a commercially available ADC module, AD8232, from Texas Instruments (Dallas, TX, USA), a commonly used block to measure, filter and amplify biopotential signals such as ECG. Its size along with the ability to operate in noisy settings is desired for applications such as wearables. The module was configured in accordance with the recommendations from the data sheet for a heart rate monitor on the torso near the heart [13], p. 8. For initial testing, the implemented filter at the output of the module caused oscillations in the signal at the ECG output. The filter was removed, and the heart rate signal was observed.

The sEMG circuit was constructed using an instrumentation amplifier with the signal from the electrodes coupling directly into the inverting and non-inverting inputs of the instrumentation amplifier. The signal then travels into a high pass Sallen-Key filter with a breakpoint set to 10 Hz. The resistors and capacitors were chosen based on a Bessel response. The Bessel function was chosen to give the best phase response since the roll-off was not essential at the filter pole [12]. An additional low pass filter with a breakpoint of 4 KHz is used to remove additional unwanted harmonics. The signal passes through a decoupling capacitor and then an op-amp buffer is used at the output to add a DC offset to the signal and couple it into low impedance measurement equipment, if needed.

The circuit also includes a three axis MEMS accelerometer (ADXL335), which can be used to remove motion artifacts [14]. The circuit dimensions are 3.8 cm × 5.7 cm × 0.6 cm. The device requires significantly less space than traditional devices used for the respective signal acquisition in healthcare settings as well as related single-modal devices.

#### 2.1.3. Construction

The circuit was constructed on a 4-layer printed circuit board, see Figure 1b, which was designed and populated by a commercial vendor. Compact layouts were created using surface mount components for each of the circuits. In some cases, for the initial design, through-hole capacitors and resistors were used in case the component values needed to be adjusted. The gel electrodes that attach to the patient were interfaced with the circuit board using 3.5 mm surface mount stereo jack connectors. These were chosen to give a reliable connection between the electrodes and the circuit board. The board can be powered from a 3 V coin cell or directly from a power supply, with the power source being changed by adjusting jumpers on the board. A 3 V DC Boost converter and a −3 V DC converter on the board are used to generate the voltages needed to power each of the components and to ensure that the DC voltage is constant if the board is powered from a battery.

### 2.2. Validation of Design

All the procedures were approved by the Institutional Review Board (IRB) for human subject research at the University of Connecticut. Twenty healthy volunteers (9 males, 11 females) of ages ranging from 19 to 37 years old were enrolled in this study. No gender-related differences have been reported for the signals in question. Subjects gave consent after reviewing the subject protocol approved by the IRB.

Six total signals were simultaneously collected, three using the prototype device and three using common laboratory-scale devices, the latter used as references for comparison to ensure an accurate validation of the device. While the reference measurements taken for the EDA, ECG, and sEMG signals utilized three different devices, all were synchronized together with the signals from the protype utilizing the PowerLab 16sp module (Figure 2e) and compatible ADInstruments LabChart 7 software. As the validation study focused primarily on the performance of the device’s circuitry, an external power supply was utilized to power the protype device at ±3.0 V.

Indicated in Figure 2a, two sets of electrodes for each signal were placed on the subject before beginning data collection—one for the prototype device and one for the reference device. The skin at all electrode sites was cleaned with 70% isopropyl alcohol before placement.

The reference EDA signal was collected using an ADInstruments GSR Amp module sampled at 1 kHz (Figure 2b). Four stainless steel electrodes, two per device, were placed on the subject’s non-dominant hand around each finger except the thumb. The reference sEMG signal was collected using an ADInstruments Dual Bio amplifier (Figure 2c). Six hydrogel Ag/AgCl electrodes, three per device, were placed on the subject’s right arm. Two were placed on the side of the bicep brachii and the third was placed on the same-arm wrist (ground). The reference ECG signal was collected using a Hewlett-Packard 78354A monitor (Figure 2d). Six hydrogel Ag/AgCl electrodes, three per device, were placed in a standard three-electrode arrangement according to Einthoven’s triangle on the subject’s chest and torso region. As the main use of this device is to gather heart rate (HR), a 3-lead system is sufficient. For each device lead, one electrode was placed under the left clavicle, the second under the right clavicle equidistant from the heart, and the third on the subject’s lower left side (ground).

#### 2.2.1. Protocol

The subject self-placed the electrodes described above with guidance from the study coordinator and assistance when needed. The electrode position on applicable muscles alternated between lateral and medial placements between subjects to eliminate any location-related bias. The experiment was conducted in a quiet, moderately lit room with an ambient temperature of 26–27 °C.

To induce a range of arousal types captured by the signals in question, the subjects underwent a four-part protocol, as outlined in Table 1. The first three parts consisted of isolated tests while the fourth was a combination of the previous three to observe simultaneous arousal. Data collection began with the subject lying in a supine position on a tilt table.

Before the first test and in-between every following test, a 3 min baseline was administered to isolate the induced stress states. The first test consisted of a 3-min, digital incongruent Stroop task and was followed by a 70° Head up Tilt (HUT) stand test. While remaining in the tilted position, the third test required the subject to repeatedly flex their biceps brachii with a 10 lb dumbbell for 10 s, resting for 30 additional seconds between each contraction. The final test required the subject to repeat the same bicep contraction previously done while simultaneously being administered the same digital incongruent Stroop task in the tilted position. Provoking multi stimuli through a combination of tests, this fourth period is referred to as Combo throughout the paper.

#### 2.2.2. Battery-Powered Test Protocol

An additional protocol was developed to test aspects of our device related to its wireless capability, specifically its ability to run off battery power. After the main protocol was conducted to completion and the data collected, a similar but amended test ran in the same lab and conditions with 3 subjects. A 3V battery was inserted into the device in lieu of the external power supply previously utilized. The same reference devices were used. Over the course of 16 min, the subjects completed a 2 min digital incongruent Stroop test, 2 min bicep test, and a 2.5 min combination test, all shortened replicas of the main protocol tasks, separated with 2.5 min rest periods.

### 2.3. Signal Processing and Data Analysis

#### 2.3.1. EDA Analysis

Four temporal and spectral indices were used to compare the EDA performance of our device to the reference device [15,16]. The EDA data was individually analyzed for all four tests outlined in the protocol. Segments of 150 s were extracted from each respective baseline and test as seen in Figure 3a. The signal segments were filtered with a lowpass FIR filter (cut-off frequency 1 Hz). The resulting signals were then down-sampled to 2 Hz.

##### Time Domain Analysis

As seen in Figure 3b, the skin conductance level (SCL) indicated by the red line is a measure emphasizing slow shifts in the tonic component of EDA while the non-specific skin conductance responses (NS.SCRs) indicated by the yellow line is a count of the number of rapid changes in the phasic component of EDA caused by a rapid response of the sympathetic nervous system due to an applied stimulus. To calculate such metrics, we used a feature extraction approach based on a nonnegative sparse deconvolution algorithm (SparsEDA) which has been used to efficiently decompose EDA data and accurately display its tonic and phasic components [17]. The reported results for SCL and NS.SCRs are the sum of the extracted tonic component and the sum of the NS.SCRs, respectively, over each specific protocol window.

##### Frequency Domain Analysis

The first spectral index utilized is a time-invariant metric based on the power spectral density of frequencies between 0.045 and 0.25 Hz (EDASympn). EDASympn was found by doing a power density analysis of the filtered and down sampled EDA signal. Welch’s periodogram method was used with 50% overlap to obtain the power spectra. A Blackman window of 128 points was then applied and the fast Fourier transform was calculated for each segment. The reported value is computed as the averaged normalized power within the frequencies of interest [18].

The second metric is a time-variant approach based on a time-frequency analysis utilizing variable frequency complex demodulation incorporating signals between 0.08 and 0.24 Hz (TVSymp). TVSymp was found by first performing a variable frequency demodulation decomposition on the filtered signal followed by a reconstruction of the signal using only the components of interest. The signal was then normalized, and its instantaneous amplitude was found using a Hilbert Transform [19]. The average value of the resulting signal during the given section was used as the TVSymp metric for comparison. An analysis of TVSymp is shown in Figure 3c where the resulting TVSymp for the prototype (red) and reference (blue) are shown for both the baseline and test period of a subjects Stroop task segment.

Both metrics have proven to be reliable interpretations of EDA and sensitive to applied stress compared to temporal measures [19,20].

#### 2.3.2. EMG Analysis

Eight temporal and spectral indices were used in the sEMG, as in [21]. For the sEMG analysis, only the segments of the protocol where the bicep was intentionally stimulated were reviewed. Both devices’ data sets were filtered with a bandpass filter (4th order Butterworth with cut-off frequency 10–500 Hz). To remove the inherent power line interference, additional 4th order Butterworth notch filters were used. The resulting filtered and raw data were then manipulated to obtain the indices.

##### Time Domain Analysis

The linear envelope was computed by fully rectifying the filtered sEMG signals and down-sampling the original sampling frequency of 1 kHz 24 times to achieve a frequency smaller than 50 Hz, 41.66 Hz, as 0–50 Hz is the sampling range known to contain frequencies most indicative of human activity and in turn closer to the motion frequencies of interest [22]. The resulting signal is representative of the signal’s estimated power and, in turn, the bicep muscles’ strength as well. Since the signals were not collected from identical, equidistance locations, variations in the outputs are inevitable. In turn, as the study mainly focuses on the shared trends and morphology of the two devices, the Pearson’s correlation was therefore computed between the two device’s sEMG envelopes to test the similarity between the two signals independent of their amplitudes. Delay between the two devices was accounted for by selecting the maximum value from the autocorrelation to ensure true alignment. A computed envelope can be seen in Figure 4a for a single contraction.

The amplitude of the linear envelope was then computed by taking the mean value throughout muscle stimulation as well as the preceding baseline stages. A *t*-test was done between the baseline and test stages to verify recorded stimulation by both devices. A second *t*-test was done between the baseline stage from both devices as well as between the test stages to observe any statistical differences.

To compute the RMS values for the sEMG signals, multiple windows of 2 ms were used to divide the filtered, unrectified data [23]. The Pearson’s correlation was computed.

##### Frequency Domain Analysis

The power spectral density (PSD) was calculated for both devices using Welch’s periodogram method with 50% overlap and a 128 point Blackman window on the raw sEMG data. Again, the Pearson’s correlation coefficient was computed between the two power spectral representations.

The Signal-to-Noise Ratio (SN Ratio) for both devices was computed using the PSD of the unfiltered signals. The calculation used assumes noise with a constant power density above 400 Hz where no power derived from muscular activity resides, as seen in Figure 4b. The final SN value is the ratio of the total power in the sEMG signal to the total power in the upper range.

The Signal to Motion Ratio (SM Ratio) for both devices was computed using the PSD of the unfiltered signals. The calculation limits motion-related artifacts to a range below 20 Hz and assumes that from 0 to 20 Hz on the signals’ PSD, the true power follows a linear trend. As such, the total power produced by motion can be estimated by summing the total power that exceeds a line from the origin to the highest mean power density, effectively separating sEMG-related power from motion-related power, as outlined in Sinderby et al. [24]. The highest mean power was calculated as the largest mean spectral value within a frequency window with a width of 12.7 Hz within the range of 35–200 Hz. Such division of the PSD can be seen in Figure 4c. The final SM value is the ratio of the total power in the sEMG signal to the total power above the estimated line.

The drop in power (DP) for both devices was computed using the PSD of the unfiltered signals. This index compares the same highest mean power density that is used in the SM ratio and calculates the lowest mean power density with the same window technique. The final DP value is the ratio between the high and low values and is indicative of the relative peak of the spectral frequencies. A higher ratio is desirable, as it demonstrates greater sensitivity to the signal’s overall amplitude fluctuations.

A *t*-test was performed for the SN, SM, and DP ratios between the two devices to note any statistical differences. Any difference notes an advantage/disadvantage of one acquisition device over the other.

#### 2.3.3. ECG Analysis

The heart rate variability (HRV) series of each protocol segment data were computed from the ECG signals of both devices using a common peak detection algorithm that implements a QRS detection function through the BioSig MATLAB library [25]. The peak detection results were manually checked for missing or inappropriate peaks to ensure accurate selection for the HRV series. A clean signal example can be seen for a short period in Figure 5 where the R peak of every QRS complex is identified and saved (red dot). The heat rate is then calculated by computing the number of peaks within a given time period, displayed by the blue plot above the ECG data in the figure.

Nine temporal and spectral indices were calculated from the series to analyze the data. The ECG analysis considers data from the Stroop, HUT, and Combo tests. The Bicep Test data were excluded due to undesirable and unwanted noise caused by movement during the task. Before computing the indices, the raw data for both devices were filtered with a band pass filter (4th order elliptic with cutoff frequency 0.5–50 Hz).

##### Time Domain Analysis

The mean RR interval (meanRR) is representative of the average time between ECG waveform peaks. The standard deviation of the signal RR intervals (SDRR) and the square root of the mean of the sum of the squares of differences between sequential RR intervals (RMSSD) are derived utilizing the meanRR. The final two indices consist of the number of pairs of neighboring RR intervals that differ by 50 ms or more (RR50) and the percentage of RR intervals that differ by 50 ms or more (pRR50).

##### Frequency Domain Analysis

The frequency analysis considered four indices derived from the PSD calculated for both devices’ HRV series using Welch’s periodogram method with 50% overlap. To compute the indices, the resulting power spectra were divided into two frequency bands including the low frequency range (0.045–0.15 Hz) and the high frequency range (0.15–0.4 Hz). The power present in these two ranges was designated as LF and HF, respectively. Additionally, the ratio between the two values (LF/HF) was considered. For further analysis, the total power of the entire HRV PSD (total) was considered, incorporating both very low frequencies and very high frequencies.

## 3. Results

### 3.1. EDA Results

Table 2 shows the overall EDA response for both the baseline and active periods in the four different test periods. SCL and NS.SCRs for both the reference device and our device show similar patterns. Generally, the fluctuations in their values from the baseline to test stages are consistent. Figure 6 displays the raw EDA components of the signal across the entire protocol using the reference and our device, respectively. Notice that, as indicated by the data between the colored lines, there is an increase in the overall tonic and phasic activity of the signal during active test portions of the protocol for both devices.

EDASympn and TVSymp also demonstrate a similar pattern when comparing the output of the reference and our device. The shared trend of TVSymp between the two devices is illustrated in Figure 3b. Notably, the overall value of each index is consistent from one device to the other simultaneously showing similar standard deviations as well. It is also observed that there is shared sensitivity amongst the tests, meaning that they change from baseline to test in a similar manner. When the reference fails to note a significant change in an index from the baseline to a test, the prototype fails to as well. Comparably, the reverse is true, as in the Combo test for example, TVSymp notes a significant difference from baseline to test for both devices. The presence of such similar morphology in the outputs demonstrates the ability of the developed device to mimic the responsiveness of the reference devices.

### 3.2. EMG Results

EMG anlysis acquired from both devices are presented in Table 3. It is clear from the signals in Figure 7 that for the resting periods, our device demonstrates significantly lower noise levels in comparison to the reference device. The devices also produced a signal with nearly identical morphology during muscle contraction.

Similar insight is provided by the interchangeability analysis results in Table 3. Both devices produced amplitudes in the same range of magnitude with no significant difference between them for the contraction sections. There was a noted difference in the relaxation stage between the devices. This can be related to the lack of noise in such periods, as previously mentioned.

The computed linear envelopes, RMS value envelope, and the PSD of the sEMG signals all displayed a large correlation. The correlation for both the linear envelope and the RMS envelope is on average higher than 0.81. Due to the small size of the bicep muscle utilized and unavoidable variability in electrode placement, it is expected that there will be some difference in the signals acquired. The correlation value for the PSD of both devices of 0.950 is notably high, disregarding small variations created by electrode placement differences.

Table 3 also includes a frequency-domain analysis comparing the sEMGs from both devices. Our device produced larger index values for all three metrics with a large statistical difference occurring for the SN and SM.

### 3.3. ECG Results

The results of the quantitative analysis of the ECG signals are found in Table 4. These indices are similar in nature to the metrics disclosed by the HRV task force [26]. Only three of the four tests in the protocol are included for the ECG analysis as the bicep portion did not target HR stimulation. A portion of the calculated HR from a subject is shown in Figure 8 for both devices.

All the indices for both devices were in the same range for each respective data set. This trend is similar amongst the standard deviations for each index as well. The LF index has some greater variation than other metrics, however, none that was statistically significant. What is important to acknowledge is the shared sensitivity between BL and Test stages between the two different devices, where the reference notes a statistical difference from BL to Test, the device does as well.

### 3.4. Battery Test Results

A side-by-side comparison of results of the battery tests can be found in Figure 9. Visibly, the trends in the data are near replications of those found in the main study protocol results above. Both of the devices show strikingly similar morphologies. During the applied stimulation periods for all three stressor stages, there is an appropriate response amongst the signal(s) for that stress type.

## 4. Discussion

A compact, integrated, trimodal EDA, sEMG, and ECG data acquisition device was developed, and its performance was compared to industry-standard devices. We found that our device was comparable to those references used for all three signals when subjects were presented with a variety of tests that provoked cognitive, physical, and orthostatic stress.

The EDA performance of our device closely matched that of the reference ADInstruments GSR Amplifier in both the time and frequency domains. The temporal measures of SCL and NS.SCRs for both devices showed similar baseline-test trends across the four tests. Since they both failed repeatedly to delineate a change in activity from rest to active periods, such characteristics are inherent of the metrics and test design, not necessarily the device. The frequency-based indices have similar values and show similar baseline-test trends for both devices across the tests, revealing our device’s ability to capture all the relevant frequency components of the desired signal. For the TVSymp indices, proven to be a true characteristic of EDA sympathetic tone [19], both devices registered significant differences from rest to active periods. Shared sensitivity to changes in EDA activity further validates the device’s performance. The results of the comparative EDA analysis are similar to previous studies validating EDA wearable devices, further increasing confidence in the device’s ability [27].

The sEMG performance of the device closely matched that of the reference ADInstruments Dual Bio Amplifier when comparing measures of similarity. It simultaneously demonstrated the advantages of the device over the reference when analyzing time- and frequency-domain indices of signal quality. High envelope correlation in the range of 0.81 with small standard deviations between the two devices, combined with similar baseline-test amplitudes, shows similar temporal behavior of the device signal in comparison to the reference. Additionally, a correlation value of 0.95 for the power spectra of the signals between the two devices as well as similar DP ratios validate the device’s ability to capture all frequency components as well as the power spectral characteristics of sEMGs. Both the SN and SM Ratios for the device are significantly higher than that of the reference device, displaying an advantage of the trimodal device. Such trends are shared with studies conducted by other groups that have validated sEMG wearable devices using similar metrics and analyses [28]. Further studies are still required to fully validate the device. A more vigorous sEMG analysis would ideally include more muscles of varying sizes to eliminate any uncertainty in the application of the device.

The ECG performance of the device nearly replicated that of the reference Hewlett-Packard 78354A monitor in terms of temporal and spectral measures derived from HRV series. The meanRR indices for the two devices were almost the exact same within a range of less than 0.5, demonstrating that the device accurately collects the peak of the ECG QRS waveform. The small variation that emerged between the indices for the remaining time-domain metrics resulted from differences found in the morphology of the wave itself which slightly modified the exact location of the peak in time. As this device is designed to measure continuous heart rate rather than heart condition/performance, the morphology of the QRS waveform itself is not of concern given continuity throughout data collection.

The frequency-domain measures of ECG were also similar across both devices for all tests. The distinct increase in power from baseline to test in the HUT portion of the protocol demonstrates the trimodal device’s ability to actively respond to increases in ECG-related stimuli, just as the reference did. Additional studies by other groups have conducted similar analyses on ECG/HRV metrics to validate wearable ECG devices, providing further confidence in the validation of this trimodal device supported by the results above [29].

When the device was tested with the amended battery protocol, no difference was detected in the device’s ability to acquire the signals in question. Increases in EDA and ECG amongst all tests, and sEMG activation during those that required muscle contraction, are visible in the device output and follow the same pattern as the references. Such similarity between the device and references, regardless of the device’s power source, confirms that its battery power capability is functional. Its validation is another step towards achieving full wireless capabilities. A more rigorous, final validation study still needs to be completed once the device is fully outfitted with all wireless components including Bluetooth capability.

## 5. Conclusions

We have demonstrated with both time- and frequency-based analyses that the integrated device designed can reliably collect EDA, sEMG, and ECG. All temporal and spectral indices calculated for our device reported similar data to those derived from the reference industry-standard devices and followed similar trends. Our device consolidates the hardware that is currently used to collect the three signals of interest into one small device that performs just as well as the current gold standard modes of acquisition. While more modifications must be made to the device in the future, this validation of the circuity on the device solidifies our confidence in its ability to acquire EDA, sEMG, and ECG signals. Overall, the main advantage of our device is its ability to streamline the collection of such signals, potentially allowing for more practical applications of trimodal data collection in the future focusing on long-term chronic pain measurement and related studies.

## Figures and Tables

**Figure 1 sensors-22-08851-f001:**
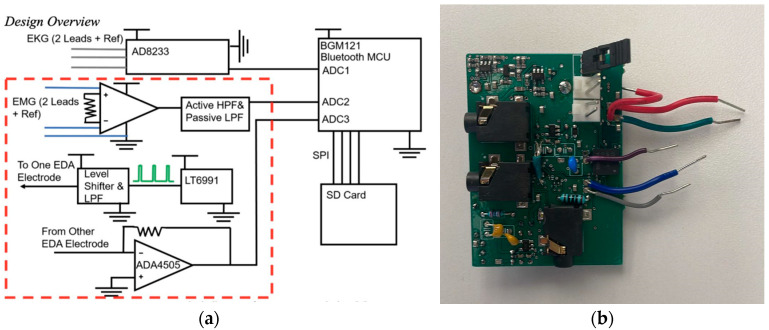
(**a**) Circuit design and (**b**) Constructed prototype chip.

**Figure 2 sensors-22-08851-f002:**
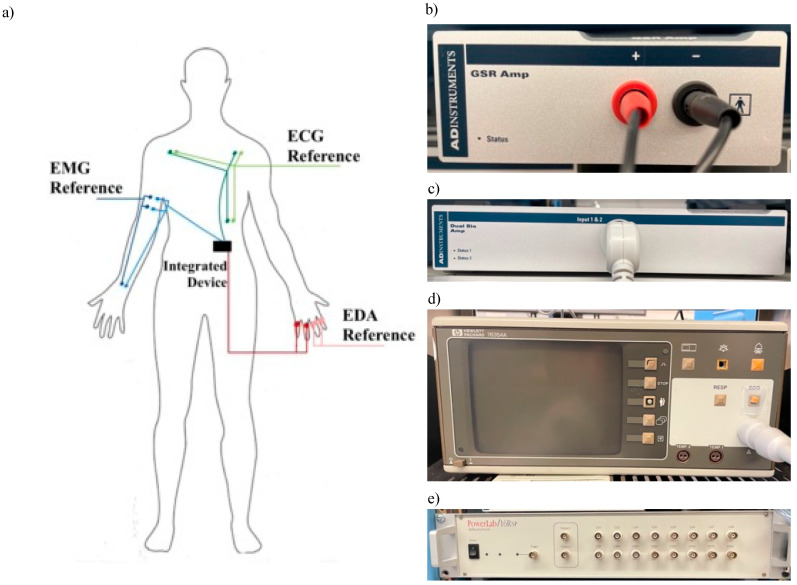
(**a**) Placement of electrodes from prototype and reference devices on subject; (**b**) ADI GSR Amp (**c**) ADI Dual Bio Amp (**d**) HP ECG Module; (**e**) PowerLab system used to acquire signals from separate modules and prototype outputs.

**Figure 3 sensors-22-08851-f003:**
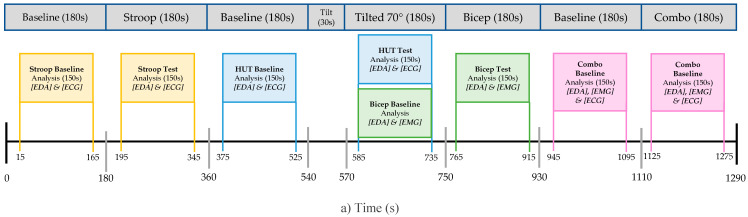
(**a**) Timeline illustrating the time epochs extracted from raw signal for each respetive analysis; (**b**) EDA time domain indices, blue (raw EDA), red (tonic SCL), yellow (NS.SCRs); (**c**) TVSymp frequency EDA analysis for prototype and reference device prior to and after stimulation.

**Figure 4 sensors-22-08851-f004:**
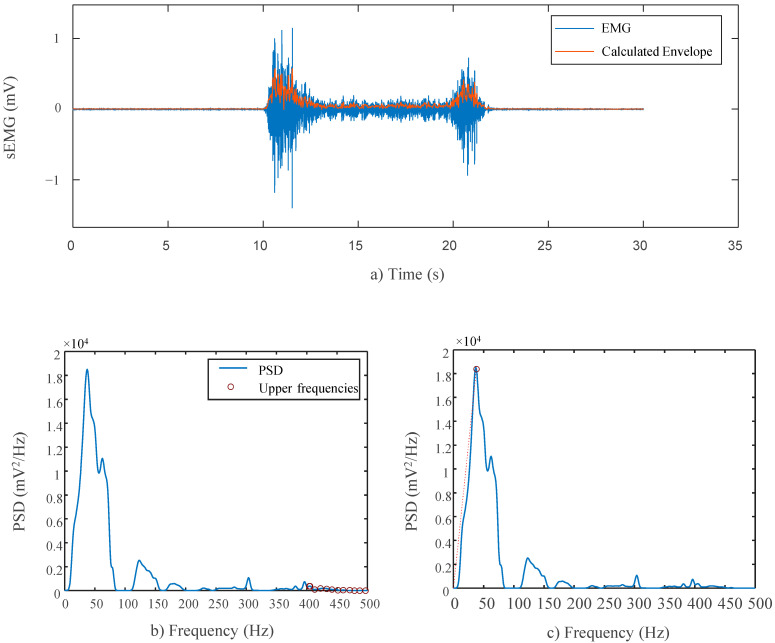
(**a**) Example of a linear envelope reconstruction—Red line displays reconstructed envelope for single unscaled EMG contraction; (**b**) SN Ratio illustration—Black circles denote high noise frequencies utilized in SN Ratio calculations; (**c**) SM Ratio illustration—Red dotted line denotes low frequency division, power above line marked as motion-related noise used in SM Ratio calculations.

**Figure 5 sensors-22-08851-f005:**
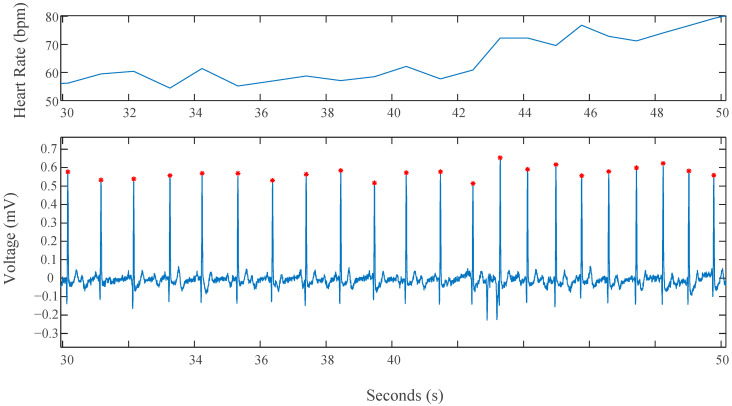
R-peak detection (**lower panel**) and results of the HR series calculated for ECG analysis (**upper panel**).

**Figure 6 sensors-22-08851-f006:**
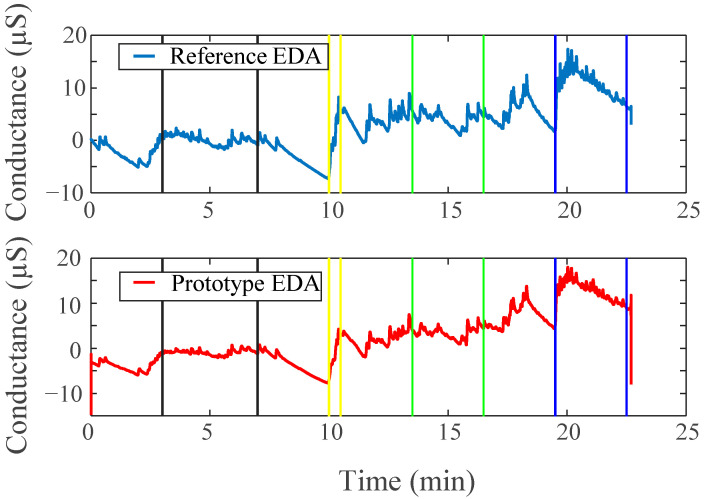
Raw EDA data from both devices, lines indicate the four different tests: black (stroop), yellow (tilt period), green (bicep), blue (combination, both stroop and bicep tests simultaneously).

**Figure 7 sensors-22-08851-f007:**
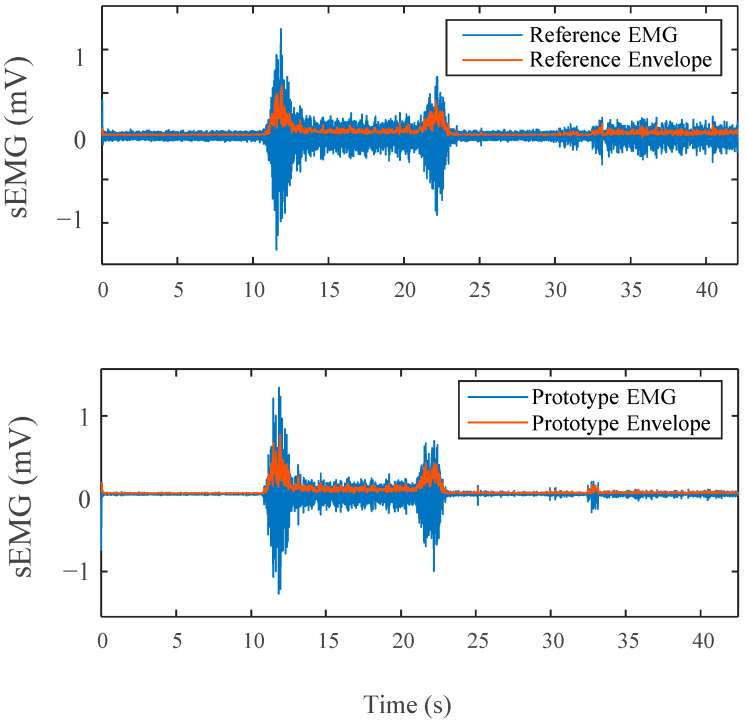
Side by Side comparison of EMG for both devices on one contraction, Envelope in Red.

**Figure 8 sensors-22-08851-f008:**
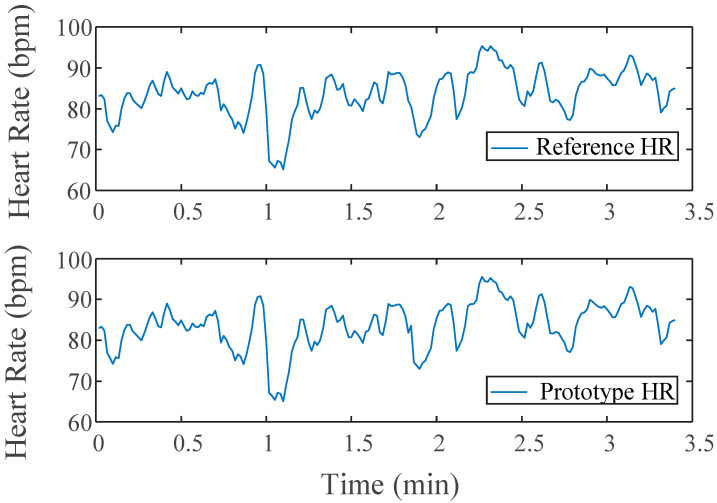
Side by Side comparison of HRV series for Prototype and Reference devices across baseline and test for Stroop Task.

**Figure 9 sensors-22-08851-f009:**
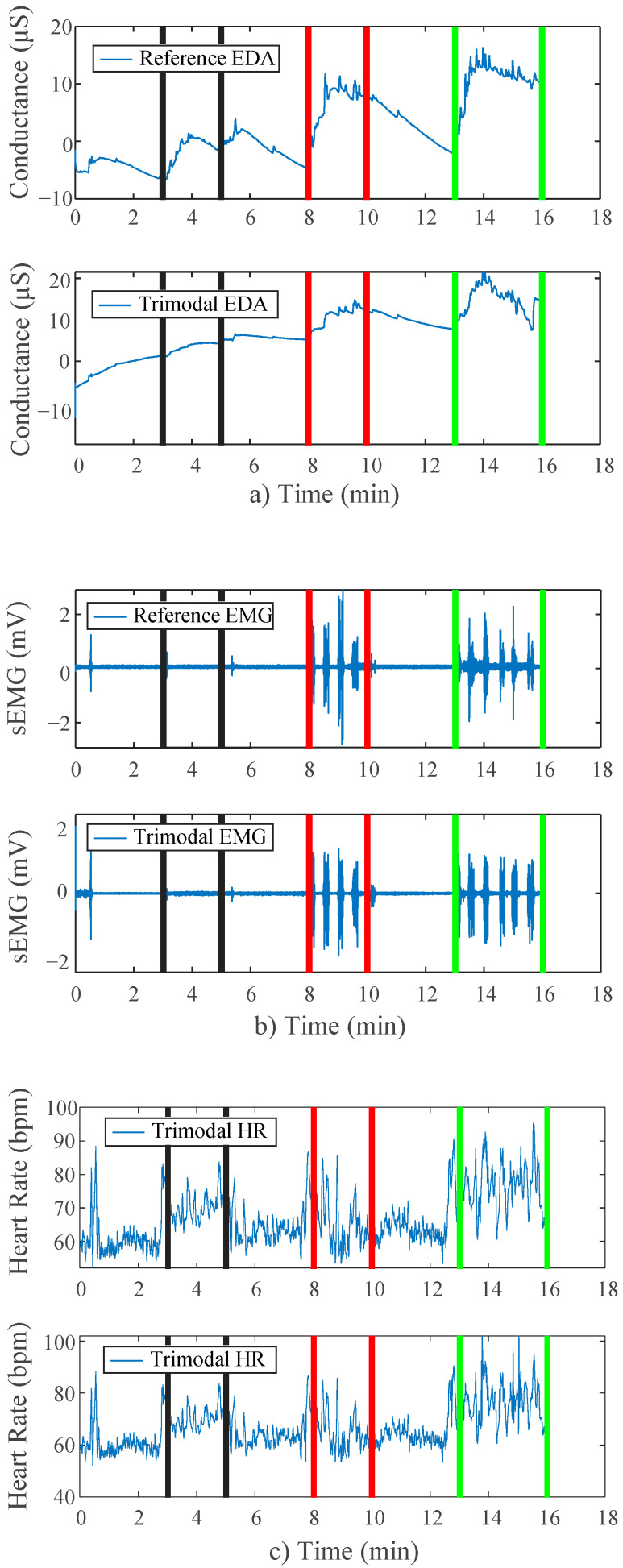
(**a**) EDA side by side comparison for modified protocol; (**b**) EMG side by side comparison for modified protocol; (**c**) HR side by side comparison for modified protocol testing battery power capability. Lines mark tests: black (Stroop), red (bicep), green (combo).

**Table 1 sensors-22-08851-t001:** Protocol outline followed.

Length (s)	Activity/Test		Remarks
180	Flat table, relaxing in supine position		Baseline
180	Stroop Task		Cognitive Stress
180	Flat table, relaxing in supine position		Baseline
30	Table Tilted		Orthostatic Stress
180	Subject in tilted position		
Repeats ×4	10	Contract and hold bicep curl	Muscular Stress
	30	Hold weight at side	
180	Tilted Table, relaxing		Baseline
Repeats ×4	10	Stroop task w/contraction	Multi Stress Combination
	30	Stroop task w/relaxation	

**Table 2 sensors-22-08851-t002:** Results for EDA Indicies.

Task	Stage	Stroop		HUT		Bicep		Combo	
		Baseline	Test	Baseline	Test	Baseline	Test	Baseline	Test
SCL	Ref	−0.92 ± 0.90	0.791 ± 1.72 *	−1.24 ± 1.38	−1.18 ± 2.14	−0.954 ± 1.45	−0.792 ± 2.3	−0.773 ± 1.6	0.526 ± 1.91
	Pro	−0.249 ± 0.67	0.0874 ± 1.27	−0.869 ± 2.0	−1.06 ± 2.09 #	−0.882 ± 1.35	−0.571 ± 2.89	−0.361 ± 1.9 #	0.26 ± 1.92
NS SCRs	Ref	1.27 ± 1.11	2.63 ± 1.24 *	1.52 ± 0.70	2.38 ± 0.87 *	2.23 ± 0.75	2.13 ± 1.12 *	1.7 ± 0.95	2.45 ± 0.84 *
	Pro	1.27 ± 0.92	2.23 ± 1.23	1.13 ± 0.76 #	2.23 ± 1.11	1.93 ± 0.88	1.73 ± 1.09 *	1.9 ± 1.47 #	2.45 ± 1.09
EDA Sympn	Ref	0.0174 ± 0.02	0.11 ± 0.22	0.0204 ± 0.04	0.282 ± 0.51 *	0.232 ± 0.38	0.143 ± 0.159 *	0.18 ± 0.44	0.282 ± 0.25 *
	Pro	0.00995 ± 0.02	0.0612 ± 0.08	0.00791 ± 0.02	0.173 ± 0.34	0.118 ± 0.21	0.142 ± 0.19	0.289 ± 0.96 #	0.292 ± 0.26
TV Symp	Ref	0.285 ± 0.16	0.422 ± 0.23	0.316 ± 0.15	0.395 ± 0.19	0.334 ± 0.18	0.322 ± 0.14	0.206 ± 0.16	0.387 ± 0.20 *
	Pro	0.302 ± 0.20	0.395 ± 0.26	0.19 ± 0.15	0.357 ± 0.22	0.285 ± 0.17	0.291 ± 0.16 *	0.23 ± 0.17	0.378 ± 0.22 *

Results for the EDA indices broken up by tests. Values are expressed as means ± SD. * Significant difference compared to baseline stage (*p* < 0.05). # Significant difference compared to Reference Device of same stage (*p* < 0.05). TVSymp, time-varying index of sympathetic skin conductance level; NS.SCRs, nonspecific skin conductance responses; EDASympn, normalized sympathetic component of the EDA; SCL, skin conductance level—Baseline, baseline reading period; Test, applied stress test period; CV, Coefficient of variation; Ref, Reference device; Pro, Prototype device.

**Table 3 sensors-22-08851-t003:** Results for EMG Indices.

			Biceps Measure
Signals Interchangeability	Amplitude Reference	BL	0.054 ± 0.038
		Test	0.133 ± 0.062 *
	Amplitude Prototype	BL	0.033 ± 0.029 #
		Test	0.109 ± 0.059 *
	Envelope Correlation		0.817 ± 0.091
	RMS Correlation		0.812 ± 0.100
	PSD Correlation		0.950 ± 0.048
sEMG indices—frequency domain			Biceps Measure
	SN ratio (dB)	Reference	41.287 ± 2.899
		Prototype	100.903 ± 1.428 #
	SM ratio (dB)	Reference	18.244 ± 7.860
		Prototype	124.939 ± 11.726 #
	DP ratio (dB)	Reference	95.934 ± 4.873
		Prototype	97.480 ± 3.650

Values are expressed as means ± SD. * Significant difference compared to baseline stage (*p* < 0.05). # Significant difference compared to Reference Device of same stage (*p* < 0.05).—BL: baseline reading period, Test: applied stress test period.

**Table 4 sensors-22-08851-t004:** Results for ECG Indices.

		Task	Stroop		HUT		Combo	
		Stage	Baseline	Test	Baseline	Test	Baseline	Test
Time Domain	MeanRR	Ref	949.42 ± 208.48	867.03 ± 181.86 *	960.40 ± 205.92	787.48 ± 173.58 *	774.08 ± 158.39	709.30 ± 158.08 *
		Dev	949.85 ± 209.54	866.99 ± 181.78 *	960.40 ± 205.99	787.41 ± 173.57 *	774.09 ± 158.41	709.19 ± 158.96 *
	SDRR	Ref	70.06 ± 38.87	59.17 ± 38.69 *	70.99 ± 36.46	57.79 ± 30.92 *	54.21 ± 31.69	47.05 ± 29.96
		Dev	67.74 ± 38.06 #	58.05 ± 39.90 *	67.79 ± 38.33	57.41 ± 31.05 *	53.83 ± 31.87	48.24 ± 30.54
	RMSSD	Ref	56.79 ± 50.72	51.88 ± 33.98	57.68 ± 46.01	46.21 ± 20.53	43.87 ± 20.55	46.44 ± 16.37
		Dev	56.72 ± 50.70	51.82 ± 35.99	53.79 ± 45.97	45.83 ± 20.74	43.88 ± 20.29	46.90 ± 16.46
	RR50	Ref	44 ± 27.29	35 ± 34.09	47.58 ± 37.65	21.95 ± 26.29 *	21.42 ± 30.88	19.26 ± 28.54
		Dev	37.79 ± 27.37	29.16 ± 31.75	37 ± 27.63	19.68 ± 23.80 *	18.42 ± 26.71	24.74 ± 29.86
	pRR50	Ref	29.96 ± 21.82	21.82 ± 22.60 *	31.51 ± 23.51	12.86 ± 16.72 *	12.25 ± 18.28	10.58 ± 17.13
		Dev	26.77 ± 23.16	19.28 ± 23.16 *	26.14 ± 22.95	12.12 ± 16.46 *	11.09 ± 17.31	13.59 ± 17.84
Frequency Domain	LF	Ref	2180.49 ± 1855.81	1760.80 ± 1399.69	1914.61 ± 1295.79	3860.96 ± 3148.36 *	3438.33 ± 4792.99	3297.14 ± 2712.38
		Dev	2105.95 ± 1843.79	1744.35 ± 1380.14	1882.74 ± 1313.79	3802.77 ± 3104.71 *	3446.74 ± 4791.74	3221.85 ± 2727.80
	HF	Ref	1940.19 ± 1876.89	1829.34 ± 3010.08	1904.13 ± 1870.69	1297.98 ± 2159.09 *	1440.99 ± 2805.01	1321.33 ± 2089.03
		Dev	1794.22 ± 1872.89	1750.23 ± 3106.36	1694.29 ± 1739.26	1258.53 ± 2153.93	1399.81 ± 2813.66	1548.46 ± 2285.58
	Total	Ref	5464.91 ± 3738.89	5155.91 ± 4948.82	5621.21 ± 4455.63	7618.63 ± 6490.12	7428.67 ± 9808.54	6899.95 ± 5554.37
		Dev	5107.11 ± 3696.44	4796.43 ± 4741.82	4834.50 ± 3537.92	7348.38 ± 6277.52 *	7229.84 ± 9803.85	7337.82 ± 6524.17
	LF/HF	Ref	2.20 ± 2.42	1.89 ± 1.20	1.72 ± 1.48	5.65 ± 3.84 *	5.34 ± 5.21	5.57 ± 3.84
		Dev	2.19 ± 2.29	2.12 ± 1.44	1.77 ± 1.41	5.90 ± 4.67 *	5.55 ± 5.34	4.87 ± 3.95 #

Values are expressed as mean ± standard deviation. * Significant difference compared to baseline stage (*p* < 0.05). # Significant difference compared to Reference Device of same stage (*p* < 0.05). MeanRR: average RR interval, SDRR: standard deviation of all RR intervals, RMSSD: square root of the mean of the sum of the squares of differences between adjacent RR intervals, RR50: number of pairs of adjacent RR intervals differing by more than 50 ms in the entire recording, pRR50: RR50 divided by total count of RR intervals, LF: low frequency range (0.045–0.15 Hz), HF: high frequency range (0.15–0.4 Hz), Total: combined range (0.045–0.4 Hz), LF/HF: ratio of HF to LF—Baseline, baseline reading period; Test, applied stress test period; HUT, Heads Up Tilt Test; Ref, Reference device; Dev, our Device.

## Data Availability

The data presented in this study are available on request from the corresponding author.

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
