# Peer review of "Design and Validation of a Multimodal Wearable Device for Simultaneous Collection of Electrocardiogram, Electromyogram, and Electrodermal Activity"

_sensors, 2022, doi:10.3390/s22228851_

Round 1
Reviewer 1 Report
Please see the attached file.

Author Response
Reviewer 1
We would like to express our gratitude towards the reviewer for finding the unintentional mistakes within the paper as well as various aspects which needed clarifying. We have modified our paper following these comments and we hope that it is now suitable for publication. Included below is a point-by-point reply to each of the comments here. Please note that all changes to the text are highlighted in yellow.
Major concerns.
1- Line 44-55. Why is a multimodal device needed for this patient population? Previous monitoring methods that are based on EDA, ECG, and sEMG provide diagnostic information. Please clearly justify why a method that combines these signals should be employed. In general, the authors should clearly justify the novelty of their work especially because two out of three modules are based on available modules or online instructions (Lines 99-115).
Thank you for noting this need for further clarification. We have included more detail with respect to the need for multimodal applications and the novelty of the integrated circuity
Section 1.0 – Paragraph #2
“Such techniques that utilize simultaneous collection of multiple signals provides greater specificity with respect to isolating the true cause of stress/pain. While one signal may be sufficient, when superimposed with the unique responses of other signals, false positives can be better eliminated through advanced signal processing and analysis. Most multimodal studies either use separate devices for each signal, many of which are laboratory, large scale devices, or use wearable devices that are only capable of simultaneously collecting one or two of the aforementioned signals [8], [9]. This type of approach may hinder the ability to gather required multimodal data due to practical limitations and ultimately cannot be efficiently implemented in a long-term diagnostic setting [10].”
Section 1.0 – Paragraph #4
“While a large portion of the circuitry is based on readily available designs, the integration of the three modules on a single chip maximizes the synchronization of the signal acquisition and the opportunities for practical implementation.”
2- The Introduction section does not provide a review of relevant literature.
Thank you for noting this. We have modified the introduction and included additional review of related topics.
See Introduction – Section 1.0
“Through novel techniques in signal processing, many signals by themselves have demonstrated the ability to widen such applications. For example, previous studies indicate that the sEMG of various muscles can be a metric of visceral pain which in turn has the ability to differentiate between types of pain such as those with or without temporomandibular dis-orders [5]. Although monomodal approaches are well suited for some applications, multimodal approaches provide better diagnostic capabilities as physiological systems are not typically affected by one particular signal. In recent years it has been seen that studies utilizing multiple signals simultaneously can potentially improve the performance of the assessment models by increasing their accuracy and specificity [6]. For example, the impact of the integration of signals has been observed in a study that concluded autonomic dysfunction in visceral pain subjects can be assessed by simultaneously evaluating the sympathetic and parasympathetic functions using EDA and heart rate variability derived from ECG [7]. Others have used multimodal approaches encompassing electroencephalograms (EEG), ECG, EMG, and EDA to analyze cognitive and physical fatigue simultaneously.
Such techniques that utilize simultaneous collection of multiple signals provides greater specificity with respect to isolating the true cause of stress/pain. While one signal may be triggered for a range of stimuli, when superimposed with the unique responses of other signals, false positives can be eliminated through advanced signal processing and analysis. Most multimodal studies either utilize separate devices for each signal, many of which are laboratory, large scale devices, or use wearable devices that are only capable of simultaneously collecting one or two of the aforementioned signals [8], [9]. This type of approach may hinder the ability to gather required data due to practical limitations and ultimately cannot be efficiently implemented in a long-term diagnostic setting [10]. To the best of our knowledge, there are no existing multimodal devices available to measure EDA, ECG, and sEMG signals simultaneously. A noninvasive, integrated multi-modal device that allows for simultaneous measurement and analysis of these signals without imposing limitations on a subject could allow us to develop a better understanding of a patient’s pain levels, especially over a long period of time. “
See:
- Stehlik et al., “Continuous Wearable Monitoring Analytics Predict Heart Failure Hospitalization: The LINK-HF Multicenter Study,” Circ: Heart Failure, vol. 13, no. 3, p. e006513, Mar. 2020, doi: 10.1161/CIRCHEARTFAILURE.119.006513.
- Szyszka-Sommerfeld, M. Machoy, M. Lipski, and K. Woźniak, “The Diagnostic Value of Electromyography in Identifying Patients With Pain-Related Temporomandibular Disorders,” Front Neurol, vol. 10, p. 180, Mar. 2019, doi: 10.3389/fneur.2019.00180.
- Ghiasi, A. Greco, R. Barbieri, E. P. Scilingo, and G. Valenza, “Assessing Autonomic Function from Electrodermal Activity and Heart Rate Variability During Cold-Pressor Test and Emotional Challenge,” Sci Rep, vol. 10, no. 1, p. 5406, Dec. 2020, doi: 10.1038/s41598-020-62225-2.
3- The paragraphs started from line 66 and line 79 include information that belongs to the Materials and Methods section. Please reorganize these parts.
Thank you for finding this misplacement. We have moved the information accordingly
Section 2.1.2. – Paragraph #4
“The circuit also includes a three axis MEMS accelerometer (ADXL335), which can be used to remove motion artifacts [14]. The circuit dimensions are 3.8 cm x 5.7 cm x 0.6 cm. The device requires significantly less space than traditional devices used for the respective signal acquisition in healthcare settings as well as related single-modal devices.”
4- The Materials and Methods section lack the details of many of the signal processing methods used to investigate the signals in time and frequency domains.
Thank you for noting this pattern. We hope the response to comments 7 and 8 as well as additional information found in the respective analysis portions provide sufficient insight.
5- The reviewer found fluff information across the manuscript. Either the same information is repeated in different sections, or unnecessary wording is used. For example (line 182): “The fourth and final test …”. Based on the understanding of the reviewer, there are four tests in this study. Therefore, the fourth test is the same as the final test. This information is redundant. Another example (line 196) “Four different indices were used to compare the EDA performance of our device to the reference device. These metrics analyzed the EDA data in both time and frequency domains [13].”: These two sentences can be easily summarized in a single shorter sentence (e.g., Four temporal and spectral indices were used to …). Surprisingly, this information is again repeated in lines 203 and 209. The authors are asked to review the whole manuscript and minimize (or even better, avoid) the fluffs. These fluffs significantly affect the clarity of the manuscript.
Thank you for your comment. The paper has been removed of additional poor/repetitive wording as noted.
6- Line 141. Are the reference signals measured simultaneously with the signals from the proposed device? How are the signals synchronized? This is a major concern, especially since the reference signals are used to validate the proposed device.
Thank you for noting this. We have included details regarding the simultaneous aspect of the signal acquisition
Section 2.2. - Paragraph #2
“Six total signals were simultaneously collected, three using the prototype device and three using common laboratory-scale devices, the latter used as references for comparison to ensure an accurate validation of the device. While the reference measurements taken for the EDA, ECG, and sEMG signals utilized three different devices, all were synchronized together with the signals from the protype utilizing the PowerLab 16sp module (Figure 2e) and compatible ADInstruments LabChart software.”
7- Line 203-207. How are the two temporal indices calculated? Please provide more details by describing the methods and Figure 3.
Thank you for pointing this out. We have included details regarding the approach taken for the time domain approach and specifically connected it to figure 3.
See Section 2.3.1 - Paragraph: #2 “Time Domain Analysis”
“To calculate such metrics, we used a feature extraction approach based on a nonnega-tive sparse deconvolution algorithm (SparsEDA) which has been used to efficiently decompose EDA data and accurately display its tonic and phasic components [17]. The reported results for SCL and NS.SCRs are the sum of the extracted tonic component and the sum of the NS.SCRs respectively, over each specific protocol window.”
8- Line 209-216. The same concern about the spectral indices and Figure 3.
Thank you for pointing this out. We have included details regarding the approach taken for the frequency domain approach, and specifically connected it to figure 3.
See Section 2.3.1 – Paragraphs #3,4 “Frequency Domain Analysis”
“ Welch’s periodogram method was used with 50% overlap to obtain the power spectra. A Blackman window of 128 points was then applied and the fast Fourier transform was calculated for each segment. The reported value is computed as the averaged normalized power within the frequencies of interest [18]. “
“ TVSymp was found by first performing a variable frequency demodulation decomposition on the filtered signal followed by a reconstruction of the signal using only the desired components of interest. The signal was then normalized, and its instantaneous amplitude was found using a Hilbert Transform [19].”
9- Line 229. Where did 41.66 Hz come from?
Many thanks for this. We have included background information on the origin of this number.
See Section 2.3.2 – Paragraph #2 “Time Domain Analysis”
“The linear envelope was computed by fully rectifying the filtered sEMG signals and down-sampling the original sampling frequency of 1kHz 24 times to achieve a frequency smaller then 50 Hz, 41.66 Hz, as 0-50Hz is the sampling range known to contain frequencies most indicative of human activity and in turn closer to the motion frequencies of interest [22]”
“The linear envelope was computed by fully rectifying the filtered sEMG signals and down-sampling from 1kHz to 41.66 Hz (1000/41.66 = 24 times decimation), to achieve a range of frequencies most indicative of human activity and in turn closer to the motion frequencies of interest which are heavily concentrated below 40 Hz [22]”
10- Line 283. Details of the feature extraction methods, including ECG R-peak detection is missing.
Thank you for your comment. We have now included information regarding the ECG extraction methods.
See Section 2.3.3 – Paragraph #1
“The heart rate variability (HRV) series of each protocol segment data were computed from the ECG signals of both devices using a common peak detection algorithm that implements a QRS detection function through the BioSig MATLAB library [25]. The peak detection results were manually checked for missing or inappropriate peaks to ensure accurate selection for the HRV series. A clean signal example can be seen for a short period in Figure 5 where the R peak of every QRS complex is identified and saved (red dot). The heat rate is then calculated by computing the number of peaks within a given time period, displayed by the blue plot above the ECG data in the figure.”
11- Figure 6. The amplitude of the proposed device output is 1 order of magnitude larger than the amplitude of the reference EDA. Please comment. Similar comment on Figure 7.
Thank you for pointing out this misrepresentation. The figures included were older versions of unscaled data. The output of the EDA circuit needed to be calibrated and converted to microSiemens from mV. The sEMG signal also needed a calibration factor. Once scaled, the amplitudes are within the same range of power and match the values provided in the tables.
12- Table 2 should be revised. The information in the first column is hard to read. It seems that the significant difference between the ref and pro is not labeled properly. For example, Stroop baseline EDASympn: Ref: 0.0174±0.02 vs Pro: 0.0095±0.02
Or
Stroop test EDASympn: Ref: 0.11±0.22 vs Pro 0.0612±0.08 What is statistical power?
Thank you for the question. The table has been adjusted to more clearly show which values correspond to Ref (reference) and Pro (prototype). In the example you provided, Ref: 0.0174±0.02 vs Pro: 0.0095±0.02 are baseline values and no statistically significant difference was found. Similarly, Ref: 0.11±0.22 vs Pro 0.0612±0.08 are Test (Stroop) values and no statistically significant difference was found. Only three Ref vs. Pro significant differences were found (marked as #): NS SCRs for HUT Test, NS SCRs for Combo baseline, and EDASymp for Combo baseline. The statistical power for the paired two-tails t-test with 20 subjects with alpha = 0.05 is 0.56.
13- Line 318. “It is also observed that there is shared sensitivity amongst the tests”. How is the sensitivity evaluated?
Thank you for the question. Sensitivity is deduced from the capability of the device to provide different values from baseline to test. The line has been reworded to improve its message.
See Section 3.1 - Paragraph #2
“It is also observed that there is shared sensitivity amongst the tests. When the reference fails to note a significant change in an index from the baseline to a test, the prototype fails to as well. Comparably, the reverse is true, as in the Combo test for example, TVSymp notes a significant difference from baseline to test for both devices. The presence of such similar morphology in the outputs demonstrates the ability of the developed device to mimic the responsiveness of the reference devices.”
14- The tables show that many of the indices are significantly different from the reference values. Please explain why the authors believe the signals are significantly similar (Line 27).
Thank you for your comment. As the data were collected using two devices (prototype and reference) simultaneously, the location of measurement is different and significant differences can arise. For this reason, most of the differences are inherent and inevitable due to the design of the procedure (non-equidistant electrodes nor identically placed electrodes). However, for most of the values, there is no significant difference. For some cases the difference is statistically significant, but they are minor in value (SCL Ref. -1.18 +/- 2.14 vs Pro. -1.06 +/- 2.09, SCL Ref. -0.773 +/- 1.6 vs. Pro. -0.361 +/- 1.9, etc.), where as in those large differences like SN ratio and SM ratio, the prototype exhibited a large advantage (SN ratio Ref. 41.287+/-2.899 vs. Pro. 100.903+/-1.428, SM ratio Ref. .18.244+/-7.860 vs. Pro. 124.939+/-11.726, etc.) For this reason we conclude that the prototype provides signals that are largely similar to the reference devices.
Minor comments.
1- Line 16. “For some approaches, a single signal is not sufficient; …”: Not sufficient for what?
Thank you for asking for clarification. We have added more detail to the sentence.
See introduction
“For some approaches, a single signal is not sufficient to provide a comprehensive diagnosis”
2- Line 18. “For instance, in visceral pain subjects the autonomic activation”: Comma is needed after the word “subjects”.
Thanks for the comment. We revised the paper accordingly.
See Introduction
“ For instance, in visceral pain subjects, the autonomic activation can be inferred using electro-dermal activity (EDA) and…”
3- Line 23. Are these signals measured simultaneously?
Thanks for asking for important clarification. We have emphasized the simultaneously nature of the collected signals
See Introduction
“This paper presents the validation of a novel multimodal low profile wearable data acquisition device for the simultaneous collection of EDA, ECG, and sEMG signals.”
4- Line 24. Are these healthy subjects?
Thanks for the comment. We revised the paper accordingly.
See Introduction
“N = 20 healthy subjects were recruited to participate in a four-stage study…”
5- Line 28. “EDA/ECG metrics showed few significant differences in trends between our device and the references.”: Not clear. Please clarify.
Thank you for asking for clarification. We have added more clarity to the sentence.
See Introduction
“Correlation of sEMG metrics ranged from 0.81 to 0.95, and EDA/ECG metrics showed few instances of significant difference in trends between our device and the references.”
6- Line 29. “With no observed differences, we demonstrated the ability of our device to collect EDA, sEMG, and ECG signals.”: This sentence contradicts the previous sentence. Please clarify.
Thank you for asking for clarification. We have fixed the contradiction noted.
See Introduction
“With only minor observed differences, we demonstrated the ability of our device to collect EDA, sEMG, and ECG signals”
7- Line 39. The very first word of the Introduction section has a typo.
Thank you for finding this unintentional typo. We have corrected the spelling.
8- Line 181. The first word appears to contain a typo.
Thank you for finding this unintentional typo. We have corrected the spelling.
9- Line 185. This information is new. It should be mentioned in the previous sections that the device used an external power supply.
Thank you for noting this. We have included power supply details earlier (see Lines 167-174)
See Section 2.2 – Paragraph #2
“As the validation study focused primarily on the performance of the device’s circuitry, an external power supply was utilized to power the protype device at +/- 3.0V.”
10- Line 203. Is this supposed to be Figure 3a rather than 2a?
Thank you for finding this unintentional typo. We have corrected the numbering.
11- Line 209. EDASympn and TVSymp should be first defined.
Thank you for noting this. We have corrected the order of the explanation.
12- Line 284. This is a minor comment. But the reviewer is curious why meanNN is used as an acronym for the mean RR interval.
Thank you for noting this confusing syntax on our part. We have clarified the spelling for all metrics previously utilizing the NN notation.
13- Line 296. The use of parentheses is not necessary for (LF) and (HF).
Thank you for finding this. We have removed the parentheses.
14- Figure 6 caption. Combo?
Thank you for pointing out this unintentional abbreviation. It has been clarified in this instance. A clearer explanation of “combo” has been introduced earlier (see lines 210-211)
See Section 2.2.1 – Paragraph #3
“Provoking multi stimuli through a combination of tests, this fourth period is referred to as Combo throughout the paper.”
15- Line 489. Reference 9 is not cited properly.
Thank you for noting this. We have edited the citation.
See Citiation no.11
“J. C. Conchell and J. Calpe, “Design Development and Evaluation of a System to Obtain Electrodermal Activity,” Analog Devices, p. 6, 2017. May 2022. [Online]. Available: https://www.analog.com/media/en/technical-documentation/tech-articles/design-development-and-evaluation-of-a-system-to-obtain-electrodermal-activity.pdf”

Reviewer 2 Report
In the manuscript “Design and Validation of a Multimodal Wearable Device for Simultaneous Collection of Electrocardiogram, Electromyogram, and Electrodermal Activity” the author “presents the validation of a novel multimodal low profile data acquisition device for EDA, ECG, and sEMG”.
First of all, I like the idea to use wearable biosensors in this combination. Maybe, you can add 3D-ACC as well – if your device is wearable at a fixed position. However, in my opinion a few questions have to be answered in addition:
1) Even if the ADC module, AD8232, should be known by experts in this field, as an author I would add a few more details to increase the readability by experts (and user) in other fields like physicians, psychologists, …
2) Specify “standard three-electrode arrangement”, please. In medicine, the common ECG derivations are Einthoven's Triangle, Goldberger and Wilson – others are commonly used but not “standard”
3) Which subset of the Stroop Task (or type) did you use (180s)?
4) I guess I overlooked it, but could you specify in more detail at which specific time epochs (of 150s) you analyzed the data in time and frequency domain, please. Maybe you can rearrange your protocol or even preferred, use a figure instead, where you add the epochs of analysis.
5) To compare signals, in my opinion the Pearson correlation is the wrong choice. Why is that: the Person correlation proofs the correlation only – but not the values. In my opinion, for a simple statistic, the Concordance-Correlation-Coefficient by Lin would be the better choice. Have a look at e.g. https://www.real-statistics.com/reliability/interrater-reliability/lins-concordance-correlation-coefficient/
6) Over all, I like developments like this one!
Author Response
Reviewer 2
We would like to express our gratitude towards the reviewer for finding the helpful comments and also noting the mistakes within the paper. We have modified our paper following these comments and we hope that it is now suitable for publication. Included below is a point-by-point reply to each of the comments here. Please note that all changes to the text are highlighted in yellow.
In the manuscript “Design and Validation of a Multimodal Wearable Device for Simultaneous Collection of Electrocardiogram, Electromyogram, and Electrodermal Activity” the author “presents the validation of a novel multimodal low profile data acquisition device for EDA, ECG, and sEMG”.
First of all, I like the idea to use wearable biosensors in this combination. Maybe, you can add 3D-ACC as well – if your device is wearable at a fixed position. However, in my opinion a few questions have to be answered in addition:
1) Even if the ADC module, AD8232, should be known by experts in this field, as an author I would add a few more details to increase the readability by experts (and user) in other fields like physicians, psychologists, …
Thank you for noting this point of. We have added additional to make the paper more widely readable.
See Section 2.1.2 – Paragraph #2
“The ECG circuit is entirely based on a commercially available ADC module, AD8232, from Texas Instruments (Dallas, TX, USA), a commonly used block to measure, filter and amplify biopotential signals such as ECG. Its size along with the ability to operate in noisy settings is desired for applications such as wearables.”
2) Specify “standard three-electrode arrangement”, please. In medicine, the common ECG derivations are Einthoven's Triangle, Goldberger and Wilson – others are commonly used but not “standard”
Thank you for your comment. We have clarified the wording to be clearer. See lines 187-188
See Section 2.2 – Paragraph #4
“Six hydrogel Ag/AgCl electrodes, three per device, were placed in a standard three-electrode arrangement according to Einthoven’s triangle on the subject’s chest and torso region.”
3) Which subset of the Stroop Task (or type) did you use (180s)?
Thank you for your question. We have specified the type of task used and its form of administration to be more. See lines 204-205.
See Section 2.2.1 – Paragraph #3
“The first test consisted of a 3-minute, digital incongruent Stroop task and was followed by…”
4) I guess I overlooked it, but could you specify in more detail at which specific time epochs (of 150s) you analyzed the data in time and frequency domain, please. Maybe you can rearrange your protocol or even preferred, use a figure instead, where you add the epochs of analysis.
Thank you for your comment. We have included a new figure along with the protocol that identifies the epochs used for the analysis.
See Section Figure 3a.
5) To compare signals, in my opinion the Pearson correlation is the wrong choice. Why is that: the Person correlation proofs the correlation only – but not the values. In my opinion, for a simple statistic, the Concordance-Correlation-Coefficient by Lin would be the better choice. Have a look at e.g. https://www.real-statistics.com/reliability/interrater-reliability/lins-concordance-correlation-coefficient/
Thank you very much for your comment. While we understand the usefulness of the CCC, we respectfully disagree that it is a better measure than Pearson’s correlation for this case. As we are collecting the signals simultaneously with two devices, those signals are collected from different locations and with different distance between electrodes and skin properties. For that reason, we cannot evaluate for concordance between the two devices, and the Parson’s correlation provides us a measure of the similarity of the trends between the devices.
We have included additional clarity in the text to clarify our decision to the reader.
See Section 2.3.2 – Paragraph 2
“Since the signals were not collected from identical, equidistance locations, variations in the outputs are inevitable. In turn, as the study mainly focuses on the shared trends and morphology of the two devices, the Pearson’s correlation was therefore computed”
Over all, I like developments like this one!
Thank you very much for your positive encouragement!

Round 2
Reviewer 2 Report
Very good revision, I have no further comments.